# Therapeutic Peptides for Treatment of Lung Diseases: Infection, Fibrosis, and Cancer

**DOI:** 10.3390/ijms24108642

**Published:** 2023-05-12

**Authors:** Shujiao Li, Yuying Li, Ying Liu, Yifan Wu, Qiuyu Wang, Lili Jin, Dianbao Zhang

**Affiliations:** 1School of Life Sciences, Liaoning University, Shenyang 110036, China; 13130271201@163.com (S.L.); yuyingli1023@163.com (Y.L.);; 2Department of Stem Cells and Regenerative Medicine, Key Laboratory of Cell Biology, National Health Commission of China, and Key Laboratory of Medical Cell Biology, Ministry of Education of China, China Medical University, Shenyang 110122, China; 2021120051@cmu.edu.cn (Y.L.); 2022120047@cmu.edu.cn (Y.W.)

**Keywords:** therapeutic peptides, antimicrobial peptide, lung disease, infection, fibrosis, lung cancer

## Abstract

Various lung diseases endanger people’s health. Side effects and pharmaceutical resistance complicate the treatment of acute lung injury, pulmonary fibrosis, and lung cancer, necessitating the development of novel treatments. Antimicrobial peptides (AMPs) are considered to serve as a viable alternative to conventional antibiotics. These peptides exhibit a broad antibacterial activity spectrum as well as immunomodulatory properties. Previous studies have shown that therapeutic peptides including AMPs had remarkable impacts on animal and cell models of acute lung injury, pulmonary fibrosis, and lung cancer. The purpose of this paper is to outline the potential curative effects and mechanisms of peptides in the three types of lung diseases mentioned above, which may be used as a therapeutic strategy in the future.

## 1. Introduction

Respiratory diseases are among the most common health problems worldwide. Many people suffer from some type of lung disease during their lifetime breathing. Most lung diseases are caused by smoking, infections, genetics, and environmental pollution. Vulnerable lungs are susceptible to acute disorders such as bacterial, viral, or fungal infections [1]. Lung infections may cause mild to severe symptoms acutely or chronically. Children, the elderly, and patients with chronic diseases are at higher risk of developing severe complications and conversion to chronic lung diseases. Chronic lung diseases, such as asthma, chronic obstructive pulmonary disease, fibrosis, and lung cancer, are a leading cause of death, accounting for 7% of all deaths worldwide. In recent years, many important advances have been made in the prevention, screening, treatment, and research of lung diseases. However, it is striking that the prevalence of chronic lung disease is still increasing, with a higher prevalence in high-income areas [2]. There is thus a critical need for the development of novel drugs and therapeutic strategies for lung diseases.

Antimicrobial peptides (AMPs), a diverse class of small peptides produced by almost all types of organisms, have an important role in the first line of antimicrobial defense. Membrane disruption, interaction with intracellular targets, immune modulation, and inflammation regulation are considered to contribute to the bioactivities of AMPs [3]. Currently, thousands of peptides have been discovered from natural sources, many of which are considered as AMPs. In addition, derivatives of natural peptides and artificial peptides have been designed and investigated in various disease models. For lung diseases, more than thirty peptides were found to be potentially beneficial for infection, fibrosis, and lung cancer. Here, the review summarized the benefits and opportunities of using therapeutic peptides for lung diseases, including infection, fibrosis, and cancer (Figure 1 and Table 1). This information has important implications for the discovery and development of peptides to intervene in lung diseases and others.

## 2. Therapeutic Peptides for Lung Infections

The lung is vulnerable to infection, which is considered the initial cause of various lung diseases. Acute infections are regularly associated with pneumonia, bronchitis, or bronchopneumonia, while chronic lung infections are often the outcome of underlying diseases. The WHO identified *P.aeruginosa*, *S.pneumoniae*, *H.influenzae,* and *S.aureus* as the four antibiotic-resistant priority infections that damage the lung in 2017 [4]. Traditional antibiotics are prescribed to treat lung diseases caused by microbial infections, while the antibiotic resistance dilemma is challenging. In recent decades, AMPs have drawn great interest from researchers due to their broad-spectrum activity, rapid lethality, and high cell selectivity [5]. For instance, Bacitracin can effectively inhibit the growth of *Streptococcus pyogenes* and *Staphylococcus aureus*, and it is currently utilized in clinical trials [6]. The cyclic lipopeptides polymyxin B and polymyxin E are used to treat Gram-negative infections, as the last line of defense against infections caused by multi-resistant Gram-negative bacteria [7].

Severe lung infection is a leading cause of acute lung injury (ALI), which is characterized by bilateral pulmonary infiltrates associated with pulmonary and non-pulmonary risk factors, as well as less severe hypoxemia. The clinical symptoms of ALI are often caused by the breakdown of the barrier function of the pulmonary capillary endothelial cells and alveolar epithelial cells [8]. Sepsis, aspiration, pneumonia, etc., are clinical risk factors for acute lung damage and acute respiratory distress syndrome, with fatality rates of 43%, 37%, 36%, and 35%, respectively [9]. Further, ALI has numerous comorbidities such as congestive heart failure, hypertension, chronic obstructive pulmonary disease, chronic renal and liver failure, sepsis, multiorgan dysfunction, and so on [10]. Lipopolysaccharide (LPS) is a component of the cell wall of Gram-negative bacteria. LPS-induced animals are commonly employed as clinically relevant ALI models [11,12]. Following LPS stimulation, nuclear factor-κB (NF-κB) signaling is triggered, to produce inflammatory cytokines and chemokines such as TNF-6, IL-6, and IL-1β to regulate the inflammatory process [13,14]. Several types of peptides presented the potential to alleviate ALI by regulating inflammatory responses and neutralizing LPS and other pathways (Table 2).

### 2.1. NF-κB Signaling Pathway

The NF-κB essential modulator (NEMO, also named IKKγ) is a subunit of IκB kinase (IKK) complex, which can phosphorylate IκB to triggers its degradation to activate NF-κB signaling [31]. The amino-terminal α-helical region of NEMO (NEMO-binding domain, NBD) was found to extenuated LPS-induced lung injury in ALI mice by inhibiting the NF-κB signaling pathway, while also inhibiting the production of pro-inflammatory cytokines and reducing oxidative stress by inhibiting NOX production [15].

Ghrelin, a 28 amino acids gastrointestinal peptide hormone, exerts a protective effect on ALI mice against sepsis via inhibiting NF-κB-iNOS pathway or Akt signaling and reducing the production of inflammatory cytokines in alveolar macrophages [16]. Ghrelin was also found to attenuate lung injury caused by acute pancreatitis in rats. The rat ALI model established by the sodium taurocholate-induced acute pancreatitis (AP) model. The ghrelin injection, either 30 min before or 3 h after AP induction, both dramatically reduced the pulmonary histological score, the water content in the lung, pulmonary microvascular permeability, MPO activity, and proinflammatory cytokines in serum. The rise of a proinflammatory mediator substance P (SP) mRNA and protein levels in the lung caused by AP were also reversed [17].

Ac2-26 is a peptide derived from the N-terminal of AnxA1. Annexin A1 (AnxA1) is an important endogenous inhibitory regulator of inflammation that has been proven to suppress NF-κB [32]. Recently, it has been reported that Ac2-26 can mitigate ischemia-reperfusion (IR)-induced lung injury. In IR lung injury, Ac2-26 significantly reduced pulmonary edema, pro-inflammatory cytokine production, oxidative stress, apoptosis, neutrophil infiltration, and lung tissue damage, as well as inhibiting NF-κB and the protein kinase pathway of mitogen activation. All of these may benefit from activating the N-formyl peptide receptor (FPR) [18].

Atrial natriuretic peptide (ANP), a kind of heart-derived secretory peptide, is a member of the natriuretic peptide family. It has been shown that ANP exerts prophylactic effects in an ALI model by reducing the induction of E-selectin expression. ANP pretreatment reduced inflammatory cell infiltration and cytokine levels in BLAF of mice with LPS-induced ALI. Moreover, ANP pretreatment significantly reduced the induction of E-selectin gene and protein levels after LPS treatment in human pulmonary artery endothelial cells and weakened the phosphorylation of NF-kB [20].

LL-37 is an antimicrobial peptide from the human cathelicidin family (amino acid sequence: LLGDFFRKSKEKIGKEFKRIVQRIKDFLRNLVPRTES) [33,34], and FF/CAP18 is an LL-37 analog known as sLL-37 (amino acid sequence: FRKSKEKIGKFFKRIVQRIFDFLRNLV). Both of them can attenuate the progression of sepsis-induced acute lung injury by inhibiting neutrophil infiltration and migration via the FAK, ERK, and P38 pathways [19]. In a mouse model of sepsis-induced acute lung injury, LL-37 and sLL-37 extenuated inflammatory responses, reduced lung injury, inhibited migration of neutrophil-like HL-60 cells, prevented neutrophil infiltration, and inactivated p-ERK, p-FAK, p-P38.

### 2.2. NLRP3 Inflammasome

Vasoactive intestinal peptide (VIP) is a neuropeptide with anti-inflammatory functions. The VIP administration alleviated lung injury (ALI) in the LPS-induced mice model by inhibiting the activation of the NLRP3 inflammasome. The VIP pretreatment suppressed the expression of NOX1 and NOX2, reduced the secretion of IL-1β and IL-18, and inhibited the activation of the NLRP3 inflammasome by restricting the NF-κB signaling pathway [21].

Liraglutide is a glucagon-like peptide (GLP-1) analog with anti-inflammatory functions in a variety of inflammatory diseases [35,36,37]. Liraglutide plays an anti-inflammatory effect on LPS-induced ALI by suppressing the NLRP3 inflammasome pathway and inhibiting the activation of Rho/NF-κB signaling in HPMECs caused by LPS [22]. Pretreatment with liraglutide reduced the histopathological damage of the lung, the wet/dry (W/D) weight ratio, protein content, and the number of inflammatory cells and proinflammatory cytokine levels in broncho-alveolar lavage fluid (BAL fluid). Further, other research shows that liraglutide reduces pulmonary inflammation by enhancing the expression of thyroid transcription factor-1 (TTF-1) and surfactant protein-A (SP-A) [23]. In LPS-induced ALI mice, liraglutide reduced lung tissue damage and restored the functional alveolar-capillary barrier. The in vitro data indicated that liraglutide restored the barrier function of endothelial monolayers through restoring intercellular connections in HPMEC, inhibited PMN-endothelial adhesion by decreasing the expression of adhesion molecules, including ICAM-1 and VCAM-1, and hence, inhibited PMN trans-endothelial migration.

### 2.3. Other Mechanisms

In LPS-induced mice models, Prdx6-PLA2 inhibitory peptides (PIPs) decrease lung injury and mouse mortality. PIP-1, PIP-2, and PIP-3 are three nonapeptides that can inhibit the phospholipase A2 activity of peroxiredoxin 6 (Prdx6) and inactivate NADPH oxidase (NOX2). All three peptides had a comparable effect on the production of ROS in mouse lungs. In particular, PIP-2 (in liposomes) inhibited ROS production similarly to NOX2 knockout. It may be claimed that NOX2-mediated ROS generation could be effectively inhibited by PIP-2 [25].

C-type natriuretic peptide (CNP) is a member of the natriuretic peptide (NP) family. CNP pretreatment markedly reduces the number of BALF cells and the concentration of inflammatory cytokines in ALI mice. The mRNA levels of MCP-1, S100A8, and E-selectin in the lung of LPS-induced mice were decreased by CNP pretreatment [26]. Thus, it was hypothesized that CNP might have a protective effect in ALI mouse models by lowering the expression of inflammatory cytokines in the lung parenchyma and the migration of lung neutrophils.

BNP is another member of the NP family. The administration of recombinant human BNP (rhBNP) suppressed LPS-induced ALI in a dog model. Pretreatment with RhBNP could minimize lung injury, pulmonary edema, the expression of IL-6 and TNF-α, and the activities of MPO and MDO in the blood [27]. Further, rhBNP was found to play protective effects against trauma-induced ALI in rat models by suppressing the JAK/STAT signaling pathway. Pretreatment with RhBNP alleviated hypoxemia and lung histopathological changes, reduced pulmonary edema and lung vascular leak, and lowered the phosphorylated protein impression of STAT1, JAK2, and STAT3 in the lung [28].

The synthetic tripeptide feG ((D-Phe)-(D-Glu)-Gly) is a novel therapeutic pharmacological agent candidate that has been proven to alleviate acute pancreatitis and acute lung injury in mouse models [38,39]. In rodent models of LPS-induced acute lung injury, the prophylactic or therapeutic administration of feG can reduce the degree of inflammatory damage by lowering leukocyte infiltration to prevent lung damage and restore lung functions [29]. Further, the prophylactic or therapeutic administration of feG was also found to prevent and ameliorate lung injury in ventilation-induced lung injury models [40].

The cyclic hexapeptide AcF (Opd-ChaWR), a C5a-C5aR signal antagonist, has been found to exhibit a protective function against cecal ligature puncture (CLP)-induced lung injury and boosts the survival rate of CLP-induced lung injury rats by suppressing C5aR activation in neutrophils [30]. Pretreatment with AcF improved diffuse pathological changes and lung edema while suppressing the proinflammatory cytokine response and neutrophil infiltration. Thus, AcF has the potential to treat severe sepsis-induced ALI.

## 3. Peptides for Lung Fibrosis

Lung fibrosis, one of the next steps of lung infections, is defined as an excessive and persistent accumulation of extracellular matrix components in response to chronic tissue damage [41]. It was also considered to be a deregulated repair process, in which the process remains permanently open, even if the classic inflammatory pathway is dampened or closed [42]. Idiopathic pulmonary fibrosis (IPF) is the most common type of interstitial lung disease and diffuse parenchymal lung disease, with no effective therapeutics. The IPF ranges from 10 to 60 cases per 100,000 individuals, and the incidence ranges from 2 to 30 cases per 100,000 individuals per year [43]. The absence of viable therapies has resulted in a 5-year survival rate of IPF circa 20% for IPF [44]. Then, the situation changed in 2014 because of the emergence of two novel anti-fibrotic agents, pirfenidone and nintedanib, which can significantly lower the risks during the progression of chronic IPF, while they might impair liver functions [45]. Although both drugs have been shown to lower mortality, about one in every five patients may experience adverse effects or diseases. For example, Pirfenidone can induce dyspepsia, anorexia, and photosensitivity, and nintedanib can induce diarrhea and nausea [46]. Peptide drugs with anti-fibrotic potential are gradually being developed (Table 3). For instance, relaxin, a member of a family of peptide hormones, can directly or indirectly suppress fibrosis through several mechanisms of action and has the potential to become a novel anti-fibrotic therapeutic agent [47].

### 3.1. NF-κB Signaling Pathway

It has been demonstrated that not just for anti-inflammation, liraglutide can also alleviate lung fibrosis in bleomycin-induced mice by inhibiting NF-κB activity which is a crucial contributor to the genesis of pulmonary fibrosis. After 28 days of intraperitoneal administration of liraglutide (2 mg/kg), inflammatory cell infiltration and TGF-β1 content in BLAF in mice were reduced, and overexpression of α-SMA and VCAM-1 were inhibited; additionally, NF-κB p65 DNA binding activity and the ratio of phosphor-NF-κB p65/total-NF-κB p65 were decreased in BLM-induced pulmonary fibrosis mice [56]. Further, pilose antler peptide (PAP), a peptide extracted and purified from velvet antler, has been reported to protect against pulmonary fibrosis by regulating the ROCK/NF-κB signaling pathway [57].

Moreover, Angiotensin 1-7 (Ang-(1-7)), a heptapeptide with anti-inflammatory and solubilizing effects, has been revealed to have preventive and therapeutic effects in bleomycin-induced pulmonary fibrosis in mice, although the therapeutic effect after modeling is much lower than that of preventive therapy, suggesting that Ang-(1-7) may not suitable for patients with ongoing fibrosis [58].

### 3.2. TGF-β Signaling Pathway

Transforming growth factor-β (TGF-β) is a crucial mediator of several fibrosis causes, including lung, rendering it a prime target for the development of anti-fibrosis drugs. The above-mentioned CNP can also specifically bind to the transmembrane guanylyl cyclase-B receptor, resulting in a substantial surge in intracellular cyclic guanosine monophosphate (cGMP) [60]. In mouse models, it was observed that CNP inhibited the TGF-β-Smad signaling pathway, reducing the transformation of the lung fibroblasts to myofibroblasts and alleviating pulmonary fibrosis [48].

Atrial natriuretic peptide (ANP), another member of the NP family, is a kind of heart-derived secretory peptide. ANP contains both anti-inflammatory and anti-fibrotic activity, and it plays a protective role in a number of organs, including the heart, blood vessels, kidneys, and lungs [20,61,62,63,64]. Okamoto, A et al. found that ANP had an anti-fibrotic and anti-inflammatory effect in BLM-induced pulmonary fibrosis via vascular endothelial cells, presumably via declining the phosphorylation of Smad2 in the TGF-β signaling pathway like the way CNP did [49].

M10 is a short peptide comprising ten amino acids (TRPASFWETS), derived from the C-terminal cytoplasmic end of the mesenchymal-epithelial transition factor (MET) chain. In accordance with ANP and CNP, the M10 peptide inhibits silica-induced fibrosis by suppressing Smad2 protein phosphorylation to prevent TGF-β1 signaling in vitro and in vivo. Further, it can reverse the EMT process induced by silica in epithelial cells and limit TGF-β1-stimulated fibroblast activation [50].

Prolyl oligopeptidase (POP) synthesizes N-acetyl-seryl-aspartyl-lysyl proline (Ac-SDKP) from its precursor thymosin-4 (T4), which is often present in organs and biological fluids [65]. As reported in a study, Ac-SDKP has a greater anti-pulmonary fibrosis capability than Tβ4 [66]. Further, it was found that Ac-SDKP co-treatment significantly reduced histological signs of BLEO-induced fibrosis inhibited TGF-β expression, as well as hindered fibroblast/myofibroblast differentiation [51].

### 3.3. RXFP1/LGR7 Receptor Pathway

CGEN25009, a novel peptide capable of binding to and activating the relaxin RXFP1/LGR7 receptor, is reported to present antifibrotic properties, including the reduction of collagen deposition and adverse bronchial remodeling, which also stimulates MMP-2 expression and reduces lung injury [59]. Therefore, CGEN25009 is expected to become a novel anti-fibrosis drug to replace relaxin due to its simple structure and inexpensive cost.

### 3.4. Oxidative Stress

The imbalance between oxidation and antioxidation is a fundamental mechanism in the development of IPF. By regulating oxidative stress and the TGF-β/MAPK signaling pathway, the peptide DHNNPQIR-NH2 (DR8), a breakdown byproduct of the natural peptide RAP (YWDHNNPQIR) which is isolated from rapeseed protein, has been demonstrated to have a protective effect against pulmonary fibrosis [52]. Likewise, fibroblast growth factor 21 (FGF21), a peptide hormone that regulates energy homeostasis, is produced by a wide range of organs [67]. FGF21 can also mitigate pulmonary fibrosis by ameliorating oxidative stress, activating the Nrf-2 pathway, and subsequently suppressing oxidative stress. This diminishes ECM deposition and pulmonary fibrosis [53].

### 3.5. Extracellular Matrix Component

Fibrosis arises due to excessive extracellular matrix component deposition, including collagen and myofibroblast, as a primary collagen-producing cell is the principal biological mediator during the period [68]. R1R2, a novel peptide derived from the bacterial adhesin SFS, has been reported to alleviate pulmonary fibrosis by regulating MMP-9 to prevent the differentiation of fibrocytes into myofibroblasts and the incursion of fibrocytes via basement membrane-like proteins [54].

Oral E4, a peptide obtained from the C-terminus of endostatin, has been observed to prevent bleomycin-induced lung fibrosis in mice by reducing the transcription of ECM components to diminish Egr-1 and ECM cross-linking by reducing LOX [55]. There is no doubt that the oral administration method is a major advantage of E4 for E4 peptide can be administered orally and exhibit an anti-fibrosis effect in view of there being few peptide and protein drugs that can be administered orally. It is worthwhile to explore and investigate its function and mechanism.

## 4. Peptides for Lung Cancer

Lung cancer, one of the last phases of lung infection or lung fibrosis, is prevalent worldwide. Lung cancer is classified into small-cell lung cancer (SCLC) and non-small-cell lung cancer (NSCLC). SCLC accounts for around 15% of all lung cancers, and surgical treatment has a dismal prognosis. Patients with stage I NSCLC have a 5-year survival rate of roughly 80%, while those in stages II to III have a survival rate of 13–60%. Surgical resection is the treatment for patients with stage I, stage II, and some stage IIIA. Other treatments, such as chemotherapy and radiation therapy, have been utilized to combat the proliferation of cancer cells [69]. However, they also interfere with the division of healthy cells, thereby impairing the recovery of normal tissue [70]. Due to the limitations of existing therapies, researchers have conducted extensive research on natural peptides and their synthetic analogs from different sources to uncover the basis of novel treatments for combating malignant cells (Table 4). Compared to other chemical agents, cationic antitumor peptides offer numerous advantages, including low molecular weight, relatively simple structure, greater cytotoxicity toward tumor cells compared to healthy cells, fewer side effects, ease of absorption, and a low risk of induced multidrug resistance [71]. The advent of natural peptide synthetic analogs, which are defined as modified or substituted variants of natural peptides, may provide a solution to the mentioned difficulties. One of the cationic antibacterial peptides, Temproin-1CEa, is a naturally occurring cationic α-helical antibacterial peptide with a potent anti-cancer effect and moderate hemolytic activity. Peptides and their analogs demonstrate considerable potential in anti-tumor drugs, with approximately 60 authorized peptide drugs on the market, four of which have sold for more than $1 billion. Among them, leuprolide acetate (Lupron; $2.12 billion), goserelin acetate (Zoladex; $1.14 billion), and octreotide acetate (Sandostatin; $1.12 billion) can be administered directly to treat cancer or certain tumor-related episodes [72]. Conceivably, peptide drugs and their analogs may also offer great potential to treat lung cancer.

Sepia ink oligopeptide (SIO), a tripeptide extracted from *Sepia esculenta*, was found to have tumor-suppressing effects [85]. To obtain peptides with higher cell permeability and stability, Zhi Zhang et.al [73] synthesized a cyclo-mimetic peptide of SIO (CSIO) by introducing SIO into disulfide bonds. The results indicate that, as compared to SIO, CSIO has a higher cellular uptake level and delivery efficiency targets lung cancer cells, inhibits the proliferation of lung cancer cells more efficiently, and induces apoptosis in lung cancer cells. Data from the Western blot and RT-PCR demonstrate that CSIO increased the expression of the apoptotic genes, caspase-3, and P53 while lowering the expression of the anti-apoptotic gene Bcl-2. According to the findings of this study, apoptosis is the primary cause of CISO’s suppression of cell proliferation. Subsequently, Xiaohua Wang et.al [74] clearly outlined the mechanism of inhibiting cell proliferation and promoting apoptosis, demonstrating that the apoptosis induction function of CISO in lung cancer cell lines is due to all three classical apoptotic pathways, including the mitochondrial pathway, the death receptor pathway, and the endoplasmic reticulum (ER)-dependent pathway.

The spider peptide toxin lycosin-I has antitumor effects to impede migration and induce apoptosis in prostate cancer cells [86]. Recent findings indicated that the anti-cancer peptide toxin LVTX-8 (IWLTALKFLGKNLGKHLAKQQLSKL) impacted lung cancer. LVTX-8 suppresses the growth and metastasis of A549 cells and H460 cells in vitro, and it suppresses the growth of the tumor by activating apoptosis and inhibits the metastasis of H460 cells and A549 cells in vivo. Transcriptomics and multiple bioinformatics analysis suggested that the reason for the suppressed function of cancer cell growth and metastasis manifested LVTX mediated related to the p53 hypoxia pathway and integrin signaling [75]. Further, the arginine modification provides an effective and modest strategy for improving the bioavailability and anticancer activity of lycosin-I, including increased cellular uptake, the improvement of intracellular distribution, and the strong toxicity produced in the 3D tumor model [87].

Cecropin is a silkworm-derived cationic lytic peptide family [88]. Cecropin B (CB), a member of the cecropin family, not only boasts a high antimicrobial activity level, but also has the ability to lyse cancer cells [88,89,90,91,92,93]. CB1a (NH2-KWKVFKKIEK-KWKVFKKIEK-AGPKWKVFKKIEK-COOH) is a CB derivative with higher selectivity and efficacy than CB. It’s surprising that lung cancer cells could be killed by CB1a in vitro, but normal cells were not affected, indicating that CB1a exhibit selective toxicity to lung cancer cells [76]. It can also hinder cancer cells from adhering together to form tumor-like spheroids. The long half-life of CB1a allows it to exert its therapeutic action in the bloodstream. According to in vivo and vitro findings, CB1a may be a promising agent for human lung cancer.

Combination therapy is one of the most important clinical treatment strategies, and peptide-peptide combination therapies have recently been employed in various diseases, such as acquired immune deficiency syndrome (AIDS) [94]. Peptide-peptide combination therapies have also been utilized to treat cancer. It has been reported that in a peptide-peptide combination, two peptides, one derived from collagen IV and the other from somatotropin, have synergized anticancer potential [95]. Recently, Cuihua Hu et.al [77] designed a peptide-peptide co-administration therapy using hybrid peptide kla-TAT and cationic anticancer peptide HPRP-A1. kla-TAT is a kla peptide that has been modified by a TAT (penetration peptide). HPRP-A1 is a 15-mer α-helical cationic peptide derived from Helicobacter pylori with strong antimicrobial and anticancer activities. Kla-TAT peptide and HPRP-A1 co-administration show synergistic effects on non-small cell lung cancer (NSCLC) A549 cells, including the promotion of mechanical disruption and apoptosis induced by a caspase-dependent pathway, and the arrest in G1 phase by the down-regulating cyclin-D1.

A previous study has demonstrated that a silk protein sericin could prevent colon carcinogenesis by minimizing oxidative stress and cell proliferation in 1,2-dimethylhydrazine-treated mice [96]. It also decreases the viability of human colorectal cancer SW480 cells in vitro [97]. Silk fibroin peptide (SFP) was obtained by removing the exterior sericin and integrating hydrolysis and enzymatic degradation methods. SFP has antiproliferative effects on lung cancer cells both in vivo and in vitro. SFP induces apoptosis in lung cancer cells by triggering cell cycle arrest in the S phase and modulating the activities of the proteins Bcl-2 and Bax, without visible damage to either healthy animals or normal lung cells [78].

According to prior research, homeobox (HOX) genes are transcriptional activators or repressors in the development of many malignancies, particularly in the aggressive proliferation of lung cancer cells [98]. The HOXA9 protein is down-regulated in lung cancer and functions as a tumor progression inhibitor [99,100], making HOXA9 a potential target for anticancer drugs. In light of that when HOXA9 is abundantly expressed in NSCLC cells, tumor aggressiveness is reduced [101], Seong-Lan Yu et al. split the protein-coding sequences of HOXA9 into three regions with overlapping parts, to determine the sequence motif of HOXA9 linked to its inhibitory action on cell motility in NSCLC cells. The results showed that the HOXA9-C fragment in NSCLC cells could limit cell invasion. The authors designed a HOXA9 active domain peptide (HADP) incorporating three sequence motifs detected in the HOXA9-C segment. Adding on, cell-penetrating peptides (CPPs) are short, non-toxic peptides. Due to their rapid cellular absorption, CPPSs have demonstrated significant promise as drug delivery agents in recent years. Therefore, Seong-Lan Yu et al. developed the CPP33-HADP recombinant protein to efficiently transport the HOXA9 protein into NSCLC. The synthetic peptide CPP33-HADP restricted the ability of A549 and NCI-H1299 cells to penetrate which gave this synthetic peptide the ability to be implemented therapeutically in metastatic lung cancer [79].

Natural killer (NK) cells, the first line of defense against disease and cancer, are crucial in controlling tumor growth. It has been shown that NK cells can effectively eliminate tumor cells that express high levels of NKG2D ligands [102]. Methionine enkephalin (MENK) is an endogenous opioid peptide with the amino acid sequence Tyr-Gly-Gly-Phe-Met. MENK has been found to enhance NK cell function and enhance the anti-tumor response [103]. Zhang et al. demonstrated that MENK reduced lung cancer cell proliferation and caused apoptosis both in vitro and in vivo. The mechanisms of MENK involved strengthening the NKG2D pathway after binding to OGFr and triggering apoptosis by modulating the OGFr/Bcl-2/Bax/caspase-3 signaling pathway. Additionally, MENK increased lung cancer cell immunogenicity by generating an ICD in tumors and improved NK cell antitumor immunity by activating the NKG2D ligand-NKG2D pathway. This study serves as a reference for the clinical management of lung cancer [80].

Peptides isolated from edible mushrooms have been proven in research to induce apoptosis in cancer cells. Arisara Prateep et al. evaluated the antitumor efficacy and underlying mechanism of a peptide extracted from the edible mushroom *Lentinus squarrosulus* in Thailand in human lung cancer cells. The findings of this study revealed that the *Lentinus squarrosulus* mushroom peptide effectively mediated lung cancer cell apoptosis by decreasing the anti-apoptotic Bcl-2 and c-FLIP proteins and raising the pro-apoptotic protein Bax [81].

SQSTM1, acting as an autophagic cargo protein, is a multi-domain protein that contributes significantly to inflammation, oncogenic transformation, autophagy, and apoptosis [104]. Yu et al. reported that the interaction between EGFR and SQSTM1 triggers SQSTM1 phosphorylation, which is a crucial signal for generating the dimerization of the SQSTM1 UBA domain and eliminating the SQSTM1 sequestration function in NSCLC. They synthesized the therapeutic peptide SAH-EJ2 which is derived from the EGFR, to block not only the binding of the EGFR to SQSTM1 but also the production of EGFR dimers. It was discovered that SAH-EJ2 could relieve the inhibitory effect of EGFR on autophagy by inhibiting Beclin1 phosphorylation, promoting EGFR degradation and limiting the activity of its downstream signal pathway, and restoring the cargo function of SQSTM1 [82].

Tan and associates developed a novel peptide R1-P2 with remarkable FGFR1 binding affinity. Peptide R1-P2 significantly suppressed the activities of FGFR1 and downstream signaling in A549 and NCI-H460 cells via lowering the phosphorylation of FGFR1 and ERK1/2. Peptides R1-P2 can inhibit the proliferation of A549 and NCI-H460 cells and trigger apoptosis. Moreover, studies corroborated the anti-angiogenic action of peptide R1-P2 in vitro and in vivo. These findings suggest that the novel peptide could be utilized to treat lung cancer individuals with aberrantly active FGFR1 [83].

According to the study by Chen et al., the phosphorylation of MARCKS plays a role in promoting lung cancer cell malignancy and migration. They also found that the MANS peptide, a truncated MARCKS N-terminal segment, antagonizes the function of MARCKS. The findings demonstrate that the MANS peptide suppresses MARCKS phosphorylation, to affects the AKT/Slug axis signal pathway, and ultimately decreases lung cancer cell motility, invasion, and metastasis [84].

## 5. Concluding Remarks

In this review, we summarized the findings concerning the roles of peptides in lung disease. As mentioned above, there are also several peptides and analogs derived from natural peptides to treat lung infection, fibrosis, and cancer. The drug exploration of lung infection, fibrosis, and cancer can focus on NF-κB signaling, and the drug and treatment developments of lung fibrosis and cancer can locate on the EMT pathway. There is a possibility to create a clinical path for lung disease treatments under the intervention of peptides in the future. A lot of research on therapeutic peptides needs to be done.

So far, over 5000 distinct therapeutic peptides have been identified [105]. They have a diverse range of structures and induce antibacterial effects while reducing the like hood of drug resistance [106]. However, applying these natural peptides has been stymied by several challenges, including low bioavailability, potential toxicity to host cells, adverse pharmacokinetic problems, and the high cost of scale production [107,108]. Since the conformation, charge, hydrophobicity, amphiphilicity, and secondary structure of antimicrobial peptides are all closely related to their bioactivities [109], we may employ a variety of methods to modify antimicrobial peptides, including molecular modification, innovative design, and hybrid design [110] to provide more possibilities and a basis for the development of antibacterial peptides with various functions, as well as more possible treatment alternatives for lung diseases.

The feasibility of delivering proteins via different routes has been investigated, including formulations suitable for oral, nasal, ophthalmic, pulmonary, buccal, and transdermal administration [111]. Currently, insulin, calcitonin, parathyroid hormone, and vasopressin are among the oral peptides in clinical studies [112]. The science of drug delivery is rapidly advancing, generating prospects for the development of peptide drugs. In addition, polypeptide-based supramolecular structures, such as hydrogels, micelles, vesicles, hybrid nanoparticles, and complexes with nucleic acids can be designed [113]. Polypeptides are utilized in a variety of biological applications, including medication administration, gene delivery, and tissue engineering. Several peptide-based nanoprobes (PBNs) have been developed that can identify the location and expression level of aberrant biomolecule activity at a diseased site, allowing for early stage diagnosis of related diseases [114]. As effective control of the various lung diseases remains a challenge, some of the data discussed may be useful guidance in the development of revolutionary therapeutic approaches for the treatment of lung disorders.

## Figures and Tables

**Figure 1 ijms-24-08642-f001:**
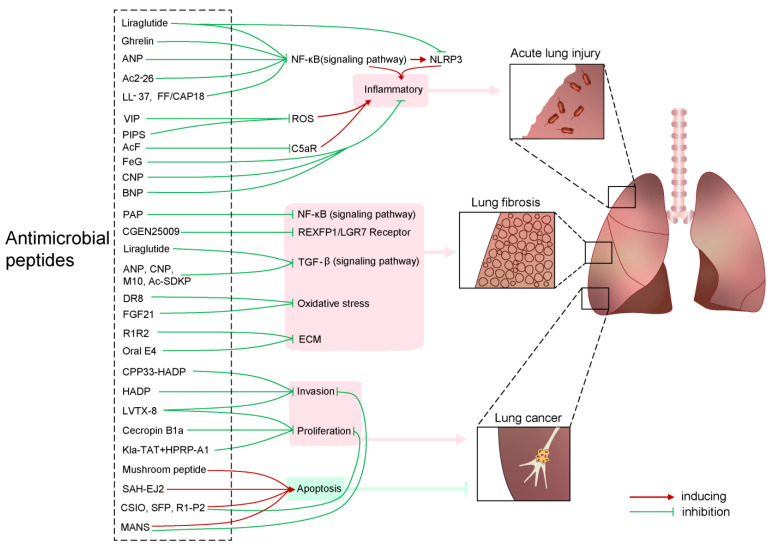
Therapeutic peptides for lung diseases. The peptides Liraglutide, Ghrelin, ANP, Ac2-26, LL-37, FF/CAP18, VIP, PIPS, AcF, FeG, CNP, and BNP inhibit acute lung injury by suppressing the NF-κB signaling pathway and suppress the activities of C5aR and ROS during the progression of inflammation. The peptides PAP, CGEN25009, Liraglutide, ANP, CNP, M10, Ac-SDKP, DR8, FGD21, and Oral E4 play antagonizing roles against the NF-κB signaling pathway, TGF-β signaling pathway, oxidative stress, the vitality of REXFP1/LGR7 receptor and the remodeling of extracellular matrix component in the progression of pulmonary fibrosis. In addition, the peptides CPP33-HADP, HADP, LVTX-8, Cecropin B1a, Kla-TAT+HPRP-A1, Mushroom peptide, SAH-EJ2, CSIO, SFP, R1-P2, and MANS suppress the development of lung cancer by inhibiting the invasion, proliferation, and promoting apoptosis of lung cancer cells.

**Table 1 ijms-24-08642-t001:** The amino acid sequences of therapeutic peptides for lung diseases.

Peptide	Sequence
Liraglutide	HAEGTFTSDVSSYLEGQAAXEFIAWLVRGRG
Ghrelin	GSSFLSPEHQRVQQRKESKKPPAKLQPR
ANP	SLRRSSCFGGRMDRIGAQSGLGCNSFRY
Ac2-26	AMVSEFLKQAWFIENEEQEYVQTVK
LL-37	LLGDFFRKSKEKIGKEFKRIVQRIKDFLRNLVPRTES
FF/CAP18	FRKSKEKIGKFFKRIVQRIFDFLRNLV
VIP	HSDAVFTDNYTRLRKQMAVKKYLNSILN
PIP-1	LYEIKHQIL
PIP-2	LHDFRHQIL
PIP-3	LYDIRHQIL
AcF	F-ornithine-PAWR
FeG	FEG
CNP	GLSKGCFGLKLDRIGSMSGLGC
BNP	SPKMVQGSGCFGRKMDRISSSSGLGCKVLRRH
CGEN25009	GQKGQVGPPGAAVRRAYAAFSVGRRAYAAFSV
M10	TRPASFWETS
Ac-SDKP	SDKP
DR8	DHNNPQIR
FGF21	HPIPDSSPLLQFGGQVRQRYLYTDDAQQTEAHLEIREDGTVGGAADQSPESLLQLKALKPG VIQILGVKTSRFLCQRPDGALYGSLHFDPEACSFRELLLEDGYNVYQSEAHGLPLHLPGNK SPHRDPAPRGPARFLPLPGLPPALPEPPGILAPQPPDVGSSDPLSMVGPSQGRSPSYAS
R1R2	GLNGENQKEPEQGERGEAGPPLSGLSGNNQGRPSLPGLNGENQKEPEQGERGEAGPP
Oral E4	SYCETWRTEAPSATGQASSLLGGRLLGQSAASCHHAYIVLCIENSFMT
CPP33-HADP	RLWMRWYSP RTRAYGHARSTRKKRCPSGGSTERQVKIWFQNRRMKMKKINK
HADP	HARSTRKKRCPSGGSTERQVKIWFQNRRMKMKKINK
LVTX-8	IWLTALKFLGKNLGKHLAKQQLSKL
Cecropin B1a	KWKVFKKIEK-KWKVFKKIEKAGPKWKVFKKIEK
Kla-TAT	KLAKLAKKLAKLAKGGRKKRRQRRR
HPRP-A1	FKKLKKLFSKLWNWK
SAH-EJ2	RRRHIVRKRTLRRLLQERE
CSIO	QPK
R1-P2	FHDAWPNLSKSS
Mans	GAQFSKTAAKG EAAAERPGEAAVA

**Table 2 ijms-24-08642-t002:** Peptides for lung infections and acute lung injury.

Peptide	Biological Effects	State Model/Object	Refs
NF-κB signaling pathway:			
NBD	TNF-α, IL-1β↓; IL-6↓; SOD↓; T-AOC activity↓; p-IKK, p-NF-κB, p65↓; NOX↓	Mice model	[15]
Ghrelin	TNF-α, IL-1β, IL-6↓; NF-κB, p65↓; p-IκBα↓; iNOS↓, p-Akt↓	Mice model	[16]
	TNF-α, IL-1, IL-6↓; MPO activity↓; SP↓	Rat model	[17]
Ac2-26	TNF-α↓; MDA↓; Bcl-2↑; caspase-3↓; p-p38, p-ERK, p-JNK↓; MKP-1↑; p-NF-κB, p65↓; IκΒ-α↑	Mice model, A549cell	[18]
LL-37	IL-6, IL-1β↓; ALT, AST, LDH↓; p-ERK, p-FAK, p-P38↓	Mice model	[19]
FF/CAP1 (sLL-37)			
ANP	E-selectin↓; TNF-α↓; IL-6, MCP-1, MIP-2, CINC-1↓; p-NF-kB↓	Mice model, HPAECs	[20]
NLRP3 inflammasome:			
VIP	NLRP3, caspase-1, IL-1β, IL-18↓; TNF-α, IL-17A↓; NOX1, NOX2↓	Mice model, primary peritoneal macrophages	[21]
Liraglutide	IL-1β, IL-18↓; MPO activity↓; wet/dry weight ratio↓; NLRP3, ASC, Caspase-1↓	Mice model	[22]
	TNF-α, IL-6, IL-1β↓; TTF-1, SP-A↑	Mice model, ATII Cell	[23]
	Rho/NF-κB signaling↓; TLR4↓; occludin, ZO-1, VE-cadherin↑; ICAM-1, VCAM-1↓	Mice model, HPMECs	[24]
Other mechanisms:			
PIP-2	ROS↓; wet/dry weight ratio↓; cells, BALF↓	Nox2 null mice	[25]
CNP	TNF-α, MIP-2, IL-6, MCP-1, KC↓; S100A8, E-selectin↓	Mice model	[26]
RhBNP	IL-6, TNL-α↓; MPO, MDA activity↓	Dog model	[27]
	JAK/STAT pathway↓; STAT1, p-STAT1↓; p-JAK2, STAT3↓	Rat model	[28]
feG	MPO activity, BAL neutrophil infiltration↓	Mice model	[29]
AcF	MPO activity, TNF-α, IL-6, IL-1β, MIP-2↓	Mice model	[30]

NBD: NEMO-binding domain peptide; VIP: Vasoactive intestinal peptide; PIP-2: Prdx6-PLA2 inhibitory peptide2; CNP: C-type natriuretic peptide; HPAECs: Human pulmonary artery endothelial cells; feG: (D-Phe)-(D-Glu)-Gly; RhBNP: Recombinant human B-type natriuretic peptide. ↑ indicates that the molecule is induced or the signaling is triggered; ↓ indicates that the molecule is reduced or the signaling is inhibited.

**Table 3 ijms-24-08642-t003:** Peptides for lung fibrosis.

Peptide	Biological Effects	State Model/Object	Refs
TGF-β signaling pathway:			
CNP	TGF-β/Smad2 signaling pathway↓; IL-1β, IL-6, bFGF mRNA↓; collagen 1A mRNA↓	Mice model, CNP transgenic mice, LF^hTERT^/GC-B cell	[48]
ANP	TGF-β/Smad2 signaling pathway↓; IL-6, MCP-1, TIMP1, IL-1β mRNA↓	Mice model, GC-A overexpressed mice, SVEC/GC-A cell	[49]
M10	α-SMA, collagen I, vimentin, fibronectin, CTGF↓; E-cadherin↑; p-Smad2↓	Mice model, A549, HBE, NIH-3T3, MRC-5 cells	[50]
Ac-SDKP	IL-17↓; α-SMA↓; TGF-β↓	Mice model	[51]
Oxidative stress:			
DR8	TGF-β/MAPK signaling pathway↓; ROS↓; IL-1β, IL-6, TNF-α↓; MCP-1, IL-8↓; HYP, CTGF, MMP-2↓	Mice model, A549 cell, NIH3T3 cell	[52]
FGF21	collagen deposition↓; TGF-β, Col I and α-SMA↓; E-cadherin↓; MDA↓; Nrf-2↑; activity of T-AOC and SOD↑	Mice model, A549 cell	[53]
Extracellular matrix component			
R1R2	collagen content↓; collagen type I content↓; myofibroblasts↓; SMAA↓; CXCR4↑; activity of MMP-9↓	Mice model	[54]
E4 peptide	Egr-1↓; LOX↓	Mice model	[55]
NF-κB signaling pathway:			
liraglutide	TGF-β1↓; α-SMA, VCAM-1↓; p-NF-κB, p65/total-NF-κB, p65↓; NF-κB p65 DNA binding activity↓	Mice model	[56]
PAP	IL-6, TNF-α, IL-1β↓; Rho, ROCK1, p-IκB, p-NF-κB↓	Mice model	[57]
Ang-(1-7)	total leukocytes↓; collagen deposition↓	Mice model	[58]
RXFP1/LGR7 Receptor Pathway:			
CGEN25009	collagen deposition↓; MMP-2↑	Mice model, THP-1 cell	[59]

CNP: C-type natriuretic peptide; ANP: Atrial natriuretic peptide; DR8: peptide DHNNPQIR-NH2; FGF21: Fibroblast growth factor 21; PAP: pilose antler peptide; Ang-(1-7): Angiotensin 1-7. ↑ indicates that the molecule is induced or the signaling is triggered; ↓ indicates that the molecule is reduced or the signaling is inhibited.

**Table 4 ijms-24-08642-t004:** Peptides for lung cancer.

Peptide	Biological Effects	State Model/Object	Refs
CSIO	Bcl-2↓; P53, caspase-3↑	A549 cells, H1299 cells	[73]
	Bcl-2↓; Bax, Caspase-9, CytoC↑; Drp1↓; Fas, Caspase-8, NIK↑; CHOP, GRP78↑	A549 cells	[74]
LVTX-8	FHL2, TAF1, MAPK8, ATM, NQO1↑; PLEC, BCAP31, DRNL, BIRC2, FNTA, LMNA, DSG2↓	A549 cells, H460 cells, nude mice xenograft tumor model	[75]
CB1a	kill cancer cells, disrupt tumor-like spheroids, inhibit the growth of lung tumors	Normal lung cells: WI-38, MRC-5, HEL-299 cells; NSCLC: A549, NCI-H209, NCI-H460, NCI-H520 cells; SCLC: NCI-H146 cells, nude strain mice	[76]
kla-TAT and HPRP-A1	anticancer activity and proliferation inhibition↑; membrane disruption, LDH leakage↑; Mitochondrial depolarization↑; ROS↑	A549 cells	[77]
SFP	Bcl-2↓; Bax↑	A549 cells, H460 cells, tumor-bearing BALB/C	[78]
HOXA9	CDH1↑; SNAI2↓	A549 cells, NCI-H1299 cells	[79]
MENK	OGFr↓; Bax↑; Bcl-2↓; caspase-3↑	A549 cells, LLC	[80]
Lentinus squarrosulus	Bcl-2↓; Bax↑	H460 cells, H292 cells, H23 cells	[81]
SAH-EJ2	LC3Ⅱ↑; EGFR↓	A549 cells, H460 cells, HCC827 cells, H1975 cells, nude mice xenograft tumor model	[82]
R1-P2	FGFR↓; ERK1/2↓	A549 cells, NCI-H1299 cells, nude mice xenograft tumor model	[83]
MANS	p-MARCKS↓; AKT/Slug↓; p-PI3K↓; p-AKT↓	A549 cells, CL1-0 cells, CL1-5 cells, PC9 cells, A549 cells, NCI-H292 cells	[84]

CSIO: cyclo-mimetic peptide of sepia ink oligopeptide; SFP: Silk fibroin peptide; MENK: Methionine enkephalin. ↑ indicates that the molecule is induced or the signaling is triggered; ↓ indicates that the molecule is reduced or the signaling is inhibited.

## Data Availability

This is a review article and no new data were created.

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
