# Peer review of "Therapeutic Peptides for Treatment of Lung Diseases: Infection, Fibrosis, and Cancer"

_ijms, 2023, doi:10.3390/ijms24108642_

Round 1

Reviewer 1 Report

This work “Antimicrobial Peptides for Treatment of Lung Diseases: Infection, Fibrosis, and Cancer” by Shujiao Li et al describes therapeutic potential of antimicrobial peptides in a variety of lung diseases. While the review claims that they present  a number of antimicrobial peptides in terms of infectious lung diseases, it needs clarification on antimicrobial peptides. For instance, liraglutide is considered GLP-1 agonist rather than antimicrobial peptides and no antimicrobial activity is described in the text. Ghrelin, 50 ANP, Ac2-26 are not considered as antimicrobial peptides to the reviewer. If the reviewers would like to claim them all as antimicrobial peptides, they need to provide the reference to show their antimicrobial activities. Majority of antimicrobial peptides are membrane active and LL-37 FF/CAP18 are the good examples. VIP, PIPS, AcF, FeG will also fit in the categories. What is unclear in the review is how their antimicrobial activities are related with their biological effects. The reviewer considers it will help the readers better if the authors classify the peptides with lytic antimicrobial peptides / non-lytic antimicrobial peptides / non-antimicrobial peptides that would be less confusing. Another option is to replace antimicrobial peptides with therapeutic peptides. 

Author Response

Dear Editor and reviewers,

Thank you for your careful checking and professional comments. The manuscript has been revised following these helpful suggestions. The changes were marked using built-in Track Changes in Microsoft Word and the main modifications are as following:

Reviewer 1:

This work “Antimicrobial Peptides for Treatment of Lung Diseases: Infection, Fibrosis, and Cancer” by Shujiao Li et al describes therapeutic potential of antimicrobial peptides in a variety of lung diseases. While the review claims that they present  a number of antimicrobial peptides in terms of infectious lung diseases, it needs clarification on antimicrobial peptides. For instance, liraglutide is considered GLP-1 agonist rather than antimicrobial peptides and no antimicrobial activity is described in the text. Ghrelin, 50 ANP, Ac2-26 are not considered as antimicrobial peptides to the reviewer. If the reviewers would like to claim them all as antimicrobial peptides, they need to provide the reference to show their antimicrobial activities. Majority of antimicrobial peptides are membrane active and LL-37 FF/CAP18 are the good examples. VIP, PIPS, AcF, FeG will also fit in the categories. What is unclear in the review is how their antimicrobial activities are related with their biological effects. The reviewer considers it will help the readers better if the authors classify the peptides with lytic antimicrobial peptides / non-lytic antimicrobial peptides / non-antimicrobial peptides that would be less confusing. Another option is to replace antimicrobial peptides with therapeutic peptides.

Response:

Thanks for your professional comments, which is important for improving the manuscript. Initially, we intended to summarize the advances of antimicrobial peptides for the potential treatment in lung diseases. And some other peptides were gradually included in later manuscripts without modification of "antimicrobial peptides ". As you point out, the term in the manuscript is inaccurate. Thanks to your comments, we replaced some of the "antimicrobial peptides" in the manuscript with "therapeutic peptides", to be more accurate.

Kind regards,

Dianbao Zhang

China Medical University

Reviewer 2 Report

Manuscript ID: ijms-2376440.

I thank the authors for the interesting manuscript.

During my revision, I could not find any special issues to be improved, but I have only a comment: it is less clear to me how the present manuscript really contributes to the AMP-focused scenario. In the literature, there are a lot of reviews and original articles reporting on the characteristics of such molecules122 (see PMID34769), and exploring their potential role in the treatment of various diseases. Would the authors explain better how they intend to contribute with their review?

Minor revision.

Author Response

Dear Editor and reviewers,

Thank you for your careful checking and professional comments. The manuscript has been revised following these helpful suggestions. The changes were marked using built-in Track Changes in Microsoft Word and the main modifications are as following:

Reviewer 2:

I thank the authors for the interesting manuscript.

During my revision, I could not find any special issues to be improved, but I have only a comment: it is less clear to me how the present manuscript really contributes to the AMP-focused scenario. In the literature, there are a lot of reviews and original articles reporting on the characteristics of such molecules122 (see PMID34769), and exploring their potential role in the treatment of various diseases. Would the authors explain better how they intend to contribute with their review?

Response:

Thank you for checking the manuscript and making comments. As you mentioned, there have been many articles and reviews of peptide discovery and functional studies. However, we have not seen any systematic review summarizing peptides with therapeutic potential for lung diseases. As peptide research gains increasing attention, its versatility provides more options for intervention strategy development for lung diseases. We summarized the related progresses in the hope of providing a broader perspective for investigators of lung disease and peptides. We believe that there will be more research advances of therapeutic peptides for lung disease interventions in the future, from basic to clinical.

Kind regards,

Dianbao Zhang

China Medical University

Reviewer 3 Report

1.     General comments 

In this review, the author summarized the findings concerning on the roles of peptides for lung disease. The review represents an advance in the significance of the potential curative effects and mechanisms of antimicrobial peptides in the three types of lung diseases such as acute lung injury, lung fibrosis and lung cancer, which may be used as a therapeutic strategy in the future.

2. Major revision

1) Figure 1

It is strongly recommended to show the amino acid sequences of antimicrobial peptides, from Liraglutide to MANS, listed in Figure 1.

3. Minor revision

1) Figure 1

a) Revise a misspelling Inflammtory” to “Inflammatory”.

b) Revise a misspelling “Apoptisis” to “Apoptosis”.

2) Page 5, Line 193: Revise a misspelling “stop” to “step”.

3) Page 9, Line 328: It is recommended to revise “the induction apoptosis function of CISO” to “the apoptosis induction (or inducing) function of CISO”.

Author Response

Dear Editor and reviewers,

Thank you for your careful checking and professional comments. The manuscript has been revised following these helpful suggestions. The changes were marked using built-in Track Changes in Microsoft Word and the main modifications are as following:

Reviewer 3:

  1. General comments

In this review, the author summarized the findings concerning on the roles of peptides for lung disease. The review represents an advance in the significance of the potential curative effects and mechanisms of antimicrobial peptides in the three types of lung diseases such as acute lung injury, lung fibrosis and lung cancer, which may be used as a therapeutic strategy in the future.

  1. Major revision

1) Figure 1

It is strongly recommended to show the amino acid sequences of antimicrobial peptides, from Liraglutide to MANS, listed in Figure 1.

Response:

Thank you for your comments. The manuscript has been improved following your suggestion. A table (Table 1) showing the amino acid sequences of peptides has been added to the revised manuscript.

  1. Minor revision

1) Figure 1

  1. a) Revise a misspelling “Inflammtory” to “Inflammatory”.
  2. b) Revise a misspelling “Apoptisis” to “Apoptosis”.

Response:

Thank you for your careful checking. Thanks to your comments, Figure 1 has been revised accordingly. The misspelling words have been corrected.

2) Page 5, Line 193: Revise a misspelling “stop” to “step”.

Response:

Following your comments, the word has been corrected accordingly.

3) Page 9, Line 328: It is recommended to revise “the induction apoptosis function of CISO” to “the apoptosis induction (or inducing) function of CISO”.

Response:

Thank you for your kindly comments. The sentence has been revised to “the apoptosis induction function of CISO” following your suggestion.

Kind regards,

Dianbao Zhang

China Medical University